# The Validity of a Three-Dimensional Motion Capture System and the Garmin Running Dynamics Pod in Connection with an Assessment of Ground Contact Time While Running in Place

**DOI:** 10.3390/s23167155

**Published:** 2023-08-14

**Authors:** Miha Drobnič, Nina Verdel, Hans-Christer Holmberg, Matej Supej

**Affiliations:** 1Faculty of Sport, University of Ljubljana, 1000 Ljubljana, Slovenia; 2Department of Health Sciences, Mid Sweden University, 83125 Östersund, Sweden; 3Department of Health Sciences, Luleå University of Technology, 97187 Luleå, Sweden; 4School of Kinesiology, University of British Columbia, Vancouver, BC V6T 1Z4, Canada

**Keywords:** 3D kinematics, accelerometer, force plate, IMU, inertial sensor, performance, wearables

## Abstract

A three-dimensional motion capture system (MoCap) and the Garmin Running Dynamics Pod can be utilised to monitor a variety of dynamic parameters during running. The present investigation was designed to examine the validity of these two systems for determining ground contact times while running in place by comparing the values obtained with those provided by the bilateral force plate (gold standard). Eleven subjects completed three 20-s runs in place at self-selected rates, starting slowly, continuing at an intermediate pace, and finishing rapidly. The ground contact times obtained with both systems differed significantly from the gold standard at all three rates, as well as for all the rates combined (*p* < 0.001 in all cases), with the smallest mean bias at the fastest step rate for both (11.5 ± 14.4 ms for MoCap and −81.5 ± 18.4 ms for Garmin). This algorithm was developed for the determination of ground contact times during normal running and was adapted here for the assessment of running in place by the MoCap, which could be one explanation for its lack of validity. In conclusion, the wearables developed for monitoring normal running cannot be assumed to be suitable for determining ground contact times while running in place.

## 1. Introduction

To objectively assess the effort and quality of exercise, it is necessary to monitor physiological and/or biomechanical parameters, such as ground contact time (GCT) while running, i.e., the time between the first contact of the foot with the ground and the moment the foot leaves the ground [1]. In general, GCT is inversely proportional to the running speed [2] and, thus, is also inversely proportional to the metabolic cost of running [3]. Most comparisons of athletes running at the same speed have concluded that a shorter GCT is associated with superior running economy [4,5,6], but in some cases, no connection between these two parameters has been observed [7,8]. Moreover, the relative lengths of the left and right GCTs play an important role in running the economy, with each 1% increase in asymmetry raising the metabolic cost by 3.7% [9]. In trained runners, the GCT has been reported to be approximately 140 ms at a speed of 7.8 m/s, 210 ms at 4.7 m/s [10], and 310 ms at 2.9 m/s [11]. For elite runners at a maximal sprinting speed, GCT is only 104 ms [12].

GCT can be measured in a number of ways, depending on the purpose and equipment available. The gold standard in this context is a bilateral force plate, which determines the area of ground contact above a given force threshold [13,14]. However, these are used solely in the laboratory, where only one or two running steps are usually assessed on a force plate. On the other hand, instrumented treadmills allow for an unlimited amount of data to be collected and have, therefore, become increasingly popular [15], although, to date, relatively few laboratories have access to such a setup. Moreover, insoles that detect plantar pressure can also collect more data, although usually at sampling frequencies several times slower than those that can be achieved with force plates. Furthermore, an increase in temperature and/or humidity inside the shoe can reduce the accuracy of insole measurements [16], and sensitivity during more powerful movements such as running can be significantly reduced, thereby lowering durability and repeatability [17].

Three-dimensional motion capture systems (MoCap) are being used more and more often to measure kinematic parameters under laboratory conditions and can, at least to a certain extent, replace the use of force plates [18]. The advantage of such systems is their ability to monitor the biomechanics of an individual’s motion without the need for additional equipment. Recently, several somewhat complex MoCap algorithms that differ for different individuals have been used to determine GCTs [13,14,19,20,21]. In one of these studies involving only rear-foot strikers, Hreljac and Stergiou [21] reported an error of 0.5 ms and a root mean square (RMS) of 7.5 ms; by contrast, in the report by Smith and co-workers on mostly rear-foot strikers [14], the corresponding values were 3.5 and 18.4 ms. To our knowledge, Handsaker and co-workers [22] have developed the most robust algorithm that is not dependent on artificial intelligence to date. Their method can be utilised under a variety of conditions (various types of ground contacts and running speeds) with a reported offset of −1.1 ms and a confidence interval of −18.6 to +16.4 ms.

When motion or GCTs are measured in the field rather than in the laboratory, inertial measurement unit (IMU) sensors are frequently used [23]. One such sensor currently available on the market is Garmin’s Running Dynamics Pod (Garmin Ltd., Olathe, KS, USA), which is attached to the top of sports shorts in the lower lumbar region. Since this sensor’s data are collected through a watch, it can be used anywhere [24]. Recently, Gonzales and colleagues [25] reviewed several investigations that were designed to validate IMU sensors [12,26,27,28,29,30,31], in two of which [27,29], the sensor was mounted on the subject’s pelvis. The validation of the IMU with an instrumented treadmill provided estimates of the accuracy and precision of GCT measurements that were −29 ms and 20 ms, respectively [27]. Validation with a force plate gave corresponding values of 4.6 and 12.1 ms [29]. Mounting the IMU on the pelvis was also shown to result in valid measurements [32], with a systematic bias of 13 ms compared to the instrumented treadmill.

Clearly, sensors and algorithms must be tested for validity before being used in practice. To our knowledge, this has not been conducted for Garmin’s Running Dynamics Pod. Several studies have documented the marketing of devices with exaggerated claims of accuracy not based on adequate scientific investigation [33,34].

During the COVID-19 pandemic, when the amount of time individuals engaged in physical activity decreased significantly [35,36,37,38,39], working out indoors became a viable option. Although the characterisation of indoor exercises, such as running in place, has, therefore, become more important [40], no studies have focused on GCT while running in place, in contrast to the recent systematic review of 49 articles on normal running [41]. In addition to its usefulness in connection with monitoring recreational exercise at home, the determination of the GCT can also be employed to detect asymmetries during rehabilitation of the legs. For instance, in connection with reactive strength exercises, such as the single leg drop jump, this time is significantly longer for the injured leg. Thus, this parameter is a valuable indicator during the rehab-related exercises, such as running in place, when training for a return to sporting activities [42]. Although other characteristics of running in place have been assessed with systems such as MoCap [43,44], IMU sensors [43,44], and electromyography (EMG) [45], there appears to have been no investigations on GCT during this activity.

Therefore, the present investigation was designed to determine these GCTs both in the laboratory using a MoCap based on infrared (IR) cameras and reflective markers, as well as in the field employing the Garmin Running Dynamics Pod based on IMU accelerometer sensors. An additional purpose was to compare the values obtained to those provided by the bilateral force plate. This approach allowed a relatively large set of GCTs to be compared to this gold standard.

## 2. Materials and Methods

### 2.1. Participants

The 11 healthy volunteers (age 22.8 ± 2.6 years; body mass 71.5 ± 18.3 kg; height 173.3 ± 9.7 cm; body mass index 23.4 ± 3.6) who participated were already familiar with the measurements to be made. The criteria for inclusion were the performance of exercise for at least 5 h each week, normal health with no injuries at the time of the study, and an age between 18 and 55 years. The senior and junior researchers, together with the technician, conducted the measurements, which were performed in the biomechanics laboratory (room temperature ~22 °C, humidity ~50%, time approximately middle of the day). The subjects had not eaten anything for at least two hours prior to testing, which they carried out in sports clothes, and reported that they were not taking any medication. They were instructed not to engage in intense exercise, refrain from consuming alcohol, and go to bed early on the day before the measurements. This research was pre-approved by the Regional Ethical Commission of the University of Ljubljana (033-16/2021-2), which follows the principles outlined by the World Medical Assembly Declaration of Helsinki. Moreover, all participants provided their written informed consent and agreed that their data could be used for publication.

### 2.2. Study Design

The subjects performed three consecutive 20-s bouts of running in a place separated by 5 s of standing still—the first slowly, the second at an intermediate pace, and the last rapidly. Each individual chose his own step rates, with the instruction that this rate should increase as evenly as possible from one 20-s bout to the next.

### 2.3. Equipment

A MoCap consisting of six Oqus infrared cameras (Oqus 7+, Qualisys, Gothenburg, Sweden) surrounding the subject was utilised to monitor the positions of two reflective markers, one on the anterior and the other on the posterior part of the foot. In addition, each subject was equipped with a Garmin Running Dynamics Pod (Garmin Ltd., Olathe, KS, USA), which was placed on the lower lumbar region of the body in accordance with the manufacturer’s instructions. Running in place was also performed on a bilateral force plate (S2P, Science to Practice Ltd., Ljubljana, Slovenia). Data from the MoCap were acquired using the Qualisys Track Manager (QTM, Qualisys, Gothenburg, Sweden); these values provided by the Garmin Running Dynamics Pod were collected by the corresponding app on a Forerunner 945 watch (Garmin Ltd., Olathe, KS, USA); the bilateral force plate was connected to a Dewe-43 analogue-to-digital converter which utilised DewesoftX software (Version 2022.1, Dewesoft Ltd., Trbovlje, Slovenia). These data were collected at 256, 1, and 1000 Hz, respectively.

The Garmin Running Dynamics Pod (Figure 1) measured 6 running parameters, including GCT, simultaneously [24].

### 2.4. Data Analysis

The synchronisation of the bilateral force plate with the Oqus cameras was ensured by utilising a start signal from the Qualisys Track Manager. In addition, the bilateral force plate was synchronised with the Garmin Pod using the time stamp of both devices.

In the case of the bilateral force plate, a ground reaction threshold of 20 N was used to calculate GCT [13,14] (Figure 2). The GCTs provided by the MoCap were compared to those obtained with the bilateral force plate for each and every step; however, for comparison with the Garmin Running Dynamics Pod, the GCTs provided by the bilateral force plate were averaged over one-second intervals.

To our knowledge, no algorithm for the determination of GCT using position data captured by a MoCap when running in place has yet been developed. Therefore, we adapted the most accurate approach in this study of normal running by Handsaker and colleagues [22] to our purposes. This approach involved two parts.

First, the time points at which the ground contact started and ended had to be determined. Handsaker and colleagues utilised the horizontal velocity of the foot to determine these time points, but this velocity was essentially zero when running in place. Therefore, contact was considered here to have been established when the position of the anterior marker signal reached a local maximum. The local minimum on the left side of the vertical–displacement curve then represented the onset of contact, while the local minimum on the right side was used to determine the end of the contact. In the second part of this approach, the maximal local acceleration on the left side of the vertical–displacement curve and the local jerk maximum on the right side were determined. The time between these two points is the GCT (Figure 3). This part was the same as that described by Handsaker et al. [22].

In some cases, Garmin did not provide GCT data; therefore, these data were excluded from the analysis (Figure 4).

### 2.5. Statistical Analysis

As assessed with the Kolmogorov–Smirnov test, the distributions of all data obtained were not normal, so non-parametric tests were utilised. Concurrent validity, which reflects the relationship between the data provided by a new device and/or algorithm and the gold standard, is reported. The Wilcoxon signed–rank test was used to evaluate a mean bias between devices while the limits of agreement (LoA) were calculated using the non-parametric Bland–Altman approach [46]. In all cases, a *p*-value of <0.05 was considered significant. The statistical power was found to be 100% for the comparisons between the FP and MoCap groups as well as the FP and Garmin groups. This was determined using G*Power software version 3.1.9.7 (Düsseldorf, Germany). All other statistical analyses were performed using MATLAB software (version R2020b, The MathWorks Inc., Natick, MA, USA).

## 3. Results

Table 1 shows the contact times detected by the different devices while running in place at various step rates. As can be seen clearly, the measurements provided by both the MoCap and Garmin Running Dynamics Pod at all three step rates differed significantly from those obtained with the gold standard.

In addition, the mean biases associated with the MoCap were between 11.5 ± 14.4 ms and 29.6 ± 18.2 ms and, in the case of the Garmin Pod, between −81.5 ± 18.4 ms and −119.6 ± 31.2 ms. Both of these systems demonstrated the lowest mean bias and LoA at the high-step rate and the greatest mean bias and LoA at the low step rate. The mean bias and LoA for all rates combined are shown in the Bland–Altman plots in Figure 5.

## 4. Discussion

The major finding here was that in comparison to the bilateral force plate, the MoCap, and Garmin Running Dynamics Pod did not provide valid ground contact times while running in place.

The average GCT at a medium step rate, as determined with the bilateral force plate, was 316 ms. This value is similar to that obtained by Chan-Roper and co-workers [11] during marathon runs, which at kilometre 40 of the race were 310 ms at a speed of 2.89 m/s. At the fast step rate, we found a GCT of 268 ms, which was slightly lower than the value for marathon runners at kilometre 8 (290 ms at a speed of 3.23 m/s) reported in that same study.

Compared to the gold standard, the MoCap overestimated (i.e., demonstrated positive mean bias) the GCTs under all conditions examined. This meant that the bias clearly tended to decline with the increasing step rate (from 29.6 ms at a low, 21.3 ms at a medium, and 11.5 ms at the fast step rate). Compared to the Garmin Running Dynamics Pod, these values were still 4–8 times closer to those obtained with the gold standard. One possible explanation for these differences is the fact that we employed a threshold of 20 N, whereas the threshold chosen by Handsaker and co-workers was 5 N [22]. In our case, it was not possible to utilise such a low threshold since preliminary attempts to replicate their method revealed that data filtering increased the error even further.

To avoid a potential signal bias due to filtering, we were required to choose this higher threshold, which is the same as that also utilised in numerous other investigations involving a force plate [6,25,26].

In contrast, compared to the gold standard, the Garmin Running Dynamics Pod significantly underestimated (negative mean bias) the GCTs at low, medium, and fast step rates, as well as for all three combined. In this case, the mean bias decreased as the step rate rose from low (−119.7 ms) to fast (−81.5 ms) but was still much larger than in the two studies testing the validity of a commercial sensor (Garmin HRM-Run™, Garmin Ltd., Olathe, KS, USA) against a MoCap [19] or a treadmill equipped with force sensors [47], which in both cases was, on average, only 10 ms during running on a treadmill.

As the rate of running in place increased, the limits of the agreement became lower. The LoA for the MoCap was again much lower (66.2 ms at low, 65.5 ms at medium, and 53.6 ms at fast step rate) than those associated with the data provided by the Garmin Running Dynamics Pod (111, 99.9, and 72 ms, respectively). For all step rates combined, the LoA for the MoCap was 70.4 ms, which is 9.6 ms less than the value reported by Adams and colleagues [48] in their test on the validity of the Garmin HRM-Run™ for determining GCT during treadmill running. For the Garmin Running Dynamics Pod, the corresponding LoA for all step rates combined was 111 ms, which is 31 ms more than in that same study.

In all devices designed to measure contact time, the accuracy of the sensor (including the sampling frequency) and the definition of the beginning and end of the contact time were crucial. The greater the noise, the larger the threshold used to define this beginning and end had to be. In addition, from a biomechanical perspective, the location of the sensor employed to determine contact time was also important. If an IMU was located on the foot, the signal changed profoundly upon impact since the foot then stopped almost immediately. However, if the accelerations were monitored more proximally (e.g., at the pelvis), the signal was less pronounced and corresponded more to the acceleration of the body’s center of mass. This challenge, in connection with the detection of contact time, could lead to larger errors. When using the MoCap for measurements, the foot must be close enough to the ground to allow for foot–ground contact to be assumed to have occurred, once again emphasising the importance of measurement accuracy and threshold definition. While it is possible to filter out noise, care must be taken, in this respect, when determining contact times since filters can distort the signal during rapid changes in motion (e.g., foot-to-ground contact).

In connection with normal running, the several different algorithms for determining GCT that have been developed provide values that differ by 22.4 ± 3.3 ms [20], 3.5 ± 18.2 ms [14], −1.1 ms [22] or 0.5 ms [21] from those obtained with the force platform. The last of these studies, with the greatest accuracy, considered heel strikes only, and the algorithm used was, therefore, inappropriate for running in places where forefoot strikes were more common. Therefore, the algorithm of Handsaker and colleagues [22] based on forefoot strikes was utilised here. This algorithm is considered to provide valid GCTs during normal running, but our current findings indicate that a valid algorithm for running in place remains to be developed. Although MoCaps other than Qualisys might exhibit a different validity in this connection, these differences are likely to be insignificant since several modern MoCap systems can demonstrate a similar absolute error in the marker distance of less than 0.6 mm in comparison to the reference value [49].

By contrast, as mentioned above, GCTs during normal running can be determined with good validity. A meta-analysis by Zeng and colleagues [50] reported that these GCTs could be determined using IMU sensors with moderate to excellent validity (ICC (95% CI) = 0.664 (0.354, 0.845), r (95% CI) = 0.811 (0.701, 0.881), *p* < 0.001) and excellent test–retest reliability (ICC (95% CI) = 0.954 (0.903, 0.978), *p* < 0.001). It should be emphasised that since the Garmin Running Dynamics pod was developed for monitoring normal running, it was presumably performed more accurately in connection with that activity. This indicated that the biomechanical differences between running in place and normal running were too extensive to allow the measurement of GCT using the same methodology.

One limitation of our study is that the rate of running in place was determined by the subjects themselves. In this way, they were able to reach their maximal rate, but at the same time, this approach made it impossible to evaluate the reliability of the two devices tested. In the future, we suggest that, as in the case of the Garmin Running Dynamics Pod, a specific algorithm be developed for both the MoCap and IMU-based devices for detecting contact times.

## 5. Conclusions

In conclusion, neither of the two systems examined here provided valid GCTs while running in place at any rate. Although the values provided by the MoCap were much closer to those obtained with the gold standard than those provided by the Garmin Running Dynamics Pod, a reliable algorithm for use during running in place remains to be developed. At the same time, two of the other five parameters assessed by the Garmin Running Dynamics Pod might still be suitable for analysing running in place, and, indeed, a measurement of cadence by this device in this context has recently been validated [51].

## Figures and Tables

**Figure 1 sensors-23-07155-f001:**
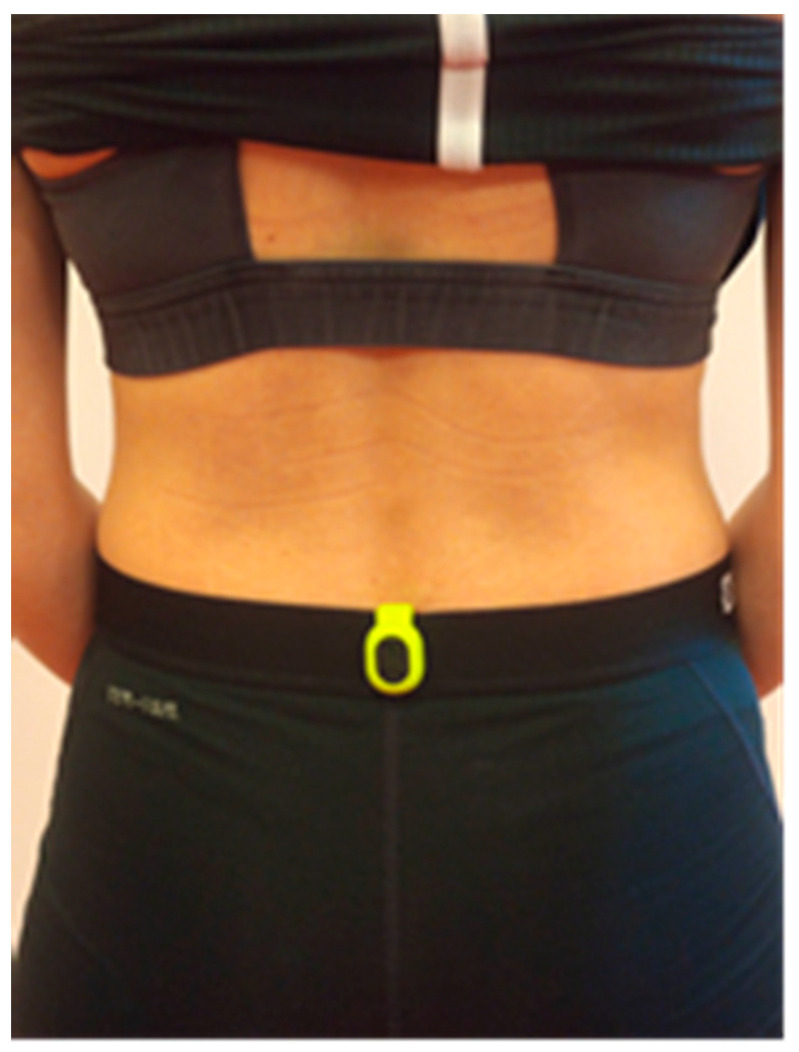
Garmin Running Dynamics Pod placement on the lower lumbar region.

**Figure 2 sensors-23-07155-f002:**
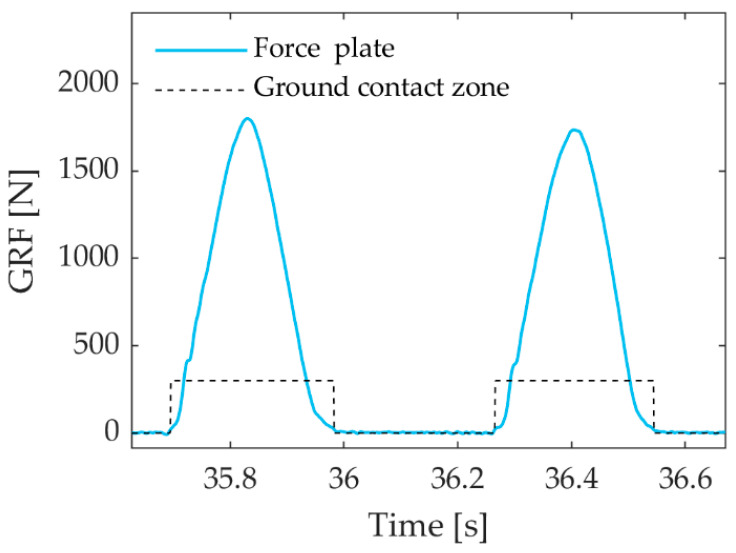
The solid line depicts the ground reaction force of the left foot on the bilateral force plate while the dashed line shows when this force was greater than 20 N, i.e., the duration of the contact of the left foot with the ground.

**Figure 3 sensors-23-07155-f003:**
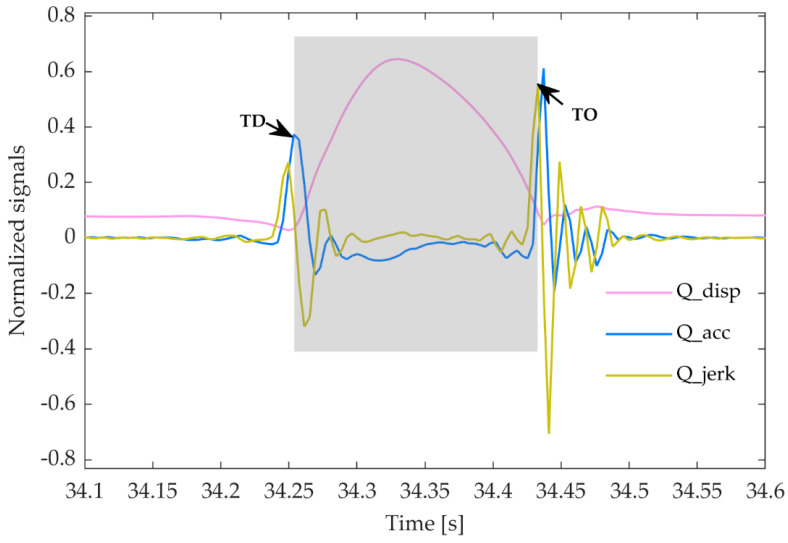
The kinematics of the marker on the anterior left foot. Each signal was normalised to its own maximal value. Q_disp—vertical displacement; Q_acc—vertical acceleration; Q_jerk—vertical jerk; TD—touchdown; TO—take-off.

**Figure 4 sensors-23-07155-f004:**
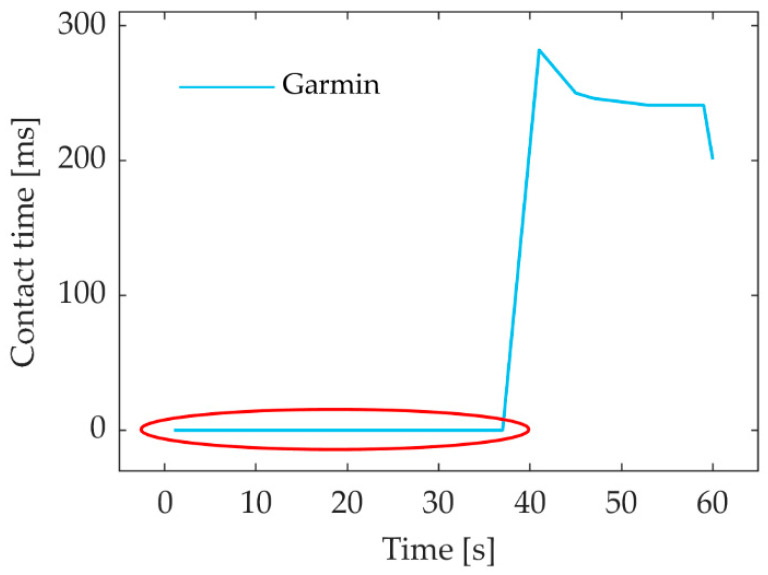
One set of GCT measurements by the Garmin Running Dynamics Pod illustrating missing data (surrounded by a red ellipse). GCT—ground contact time.

**Figure 5 sensors-23-07155-f005:**
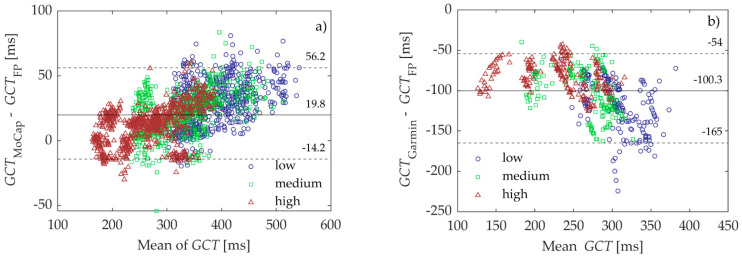
Bland–Altman plots comparing the ground contact time (GCT) determined with (**a**) The three-dimensional motion capture system (MoCap) and (**b**) the Garmin Running Dynamics Pod (Garmin) to those determined with the bilateral force plate (FP). Blue circles = low step rate, green squares = medium step rate, red triangles = fast step rate. The dashed lines represent the limits of agreement (LoA), while the solid line depicts the mean bias.

**Table 1 sensors-23-07155-t001:** Comparison of the ground contact times (GCTs) at different step rates as determined by the three-dimensional motion capture system (MoCap), the Garmin Running Dynamics Pod (Garmin) and the gold standard bilateral force plate (FP). LoA—limits of agreement.

Step Rate	FP [ms]	MoCap [ms]	Garmin [ms]	Mean Bias [ms]	*p*-Value	LoA [ms]
Low	377.1 ± 52.0	406.7 ± 59.6	/	29.6 ± 18.2	<0.001 *	−5.7 to 60.5
	368.6 ± 39.4	/	249.0 ± 27.4	−119.6 ± 31.2	<0.001 *	−184.8 to −74.8
Medium	321.9 ± 55.5	343.2 ± 64.0	/	21.3 ± 17.2	<0.001 *	−14.2 to 51.3
	316.1 ± 43.2	/	214.9 ± 32.6	−101.2 ± 28.4	<0.001 *	−155.8 to −55.9
High	255.9 ± 54.5	267.4 ± 60.9	/	11.5 ± 14.4	<0.001 *	−15.2 to 38.4
	267.8 ± 50.9	/	186.3 ± 46.7	−81.5 ± 18.4	<0.001 *	−199.8 to −47.8
All	311.6 ± 73.6	331.4 ± 84.1	/	19.8 ± 18.1	<0.001 *	−14.2 to 56.2
	316.3 ± 61.0	/	216.0 ± 44.8	−100.3 ± 30.7	<0.001 *	−165.0 to −54.0

* Significantly different mean GCT.

## Data Availability

The data presented in this study are available on request from the corresponding author provided that the access does not interfere with the conditions provided by the ethics committee.

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
