# Peer review of "The Validity of a Three-Dimensional Motion Capture System and the Garmin Running Dynamics Pod in Connection with an Assessment of Ground Contact Time While Running in Place"

_sensors, 2023, doi:10.3390/s23167155_

Round 1

Reviewer 1 Report (New Reviewer)

1. the title should be revised to represent the main idea of your manuscript.

2. the figures should be improved to enhance the readability.

3. the references should include the latest research, especially the papers from the past three years.

the English is good.

Author Response

Reviewer 2 Report (New Reviewer)

An interesting approach to a subject. The authors analyse and compare the results obtained with two methods and emphasise that it cannot be assumed that the wearables developed for monitoring normal running are suitable for determining ground contact times when running on the spot.

The authors have achieved their goal. They have also proposed a methodology (one adapted from Handsaker and colleagues) and I think it is necessary to stress whether their methodology gives more accurate results (in terms of discussions/conclusions/recommendations) compared to the methods used as reference (MoCap and the Garmin Running Dynamics Pod)?

What research method can provide accurate results regarding ground contact times when running in place? Perhaps it would also be interesting to analyse/discuss or consider the influence of the biomechanics of the human body on this topic.

Author Response

Reviewer 3 Report (New Reviewer)

Dear Authors,

I would like to express my gratitude for the opportunity to review this manuscript.

At this stage, the document requires improvements, below with line indication:

Page 1, bottom left, template is from 2022, please change it.

2-4 – Please revise the upper and lower case in the title.

6-11 – Please indicate the authors´ initials, according to the contributions (page 8).

35 – Previously abbreviated, in this line not necessary in full.

61 – “Hreljac and colleagues [21]” – This is incorrect, only two authors in the reference. Please revise this line and carefully revise all manuscript regarding the citations format.

62 – RMS should be in full.

64-72 – These two paragraphs are too small. Please consider standardizing the paragraphs size, aiming to improve readability. This should also be considered in the discussion section, for example, lines 248-253.

68 – Please describe IMU in full in the first appearance in the text.

70 – “Garmin’s Running Dynamics Pod” – please indicate the city and country of the manufacturer.

83 – Please correct the beginning of the paragraph.

94 – Please describe in full EMG.

97 - Please describe in full IR.

104 – Please include decimals.

104 – Please consider indicating MBI and % of BF.

105 – Please describe in detail the inclusion e exclusion criteria.

111 – Please describe all details related to data collection. For example, who collected the data (academic background and experience), and other details related to data collection (space characterization, time of day, temperature, humility, clothes, nutrition, medicine, training routines, refrain from intense training? These are only some examples, please consider all details.

113 – How was exercise intensity measured?

129 – Please confirm the hz values.

144 – Please improve the figure 2 quality.

167 – Please improve the figure 4 quality.

171- Previously to the results section, a statistical analysis section should be written in the manuscript. Please consider detailed information, for example regarding statistical power (Gpower).

182 – Please revise the table content.

197 – The abbreviation is previously in full in the manuscript.

213 – Please confirm in the journal template if the citation numbers are separated by spaces or only commas.

248 – Please correct the beginning of the paragraph.

260 – Please consider developing the discussion section.

260 – Please consider a final paragraph in the discussion section with study limitations and suggestions for future research.

260 – It is suggested the inclusion in the manuscript of a conclusions section.

279 - Please carefully revise the references format, they are not according to the journal´s template.

Please carefully revise the English throughout the manuscript and document format details, considering the journal template.

Moderate editing of English language required.

Round 2

Reviewer 1 Report (New Reviewer)

The authors has addressed my comments.

The writing is good.

Author Response

Reviewer 3 Report (New Reviewer)

Dear Authors,

Thank you for considering my suggestions and incorporating them into the manuscript, which is globally improved, congratulations.

Below are suggestions related to this last version (v2), with line indication.

2-4 – Please revise the title format, normally in the journal with words starting in uppercase.

121 & 126   Please standardize the subtitles format.

181 – A subtopic entitles “statistical analysis” should be added.

187-188 – Please describe all the results of statistical power and include the GPower city and country.

Please double-check all the references’ format

Please double-check all the manuscript format.

Please double-check the English throughout the manuscript.

Minor editing of English language required.

Author Response

This manuscript is a resubmission of an earlier submission. The following is a list of the peer review reports and author responses from that submission.

Round 1

Reviewer 1 Report

Its noteasy to judge the quality of this manuscripts. The author based on the Garmin Running Dynamics Pod devises evaluated two systems. The conclusions were given through data comparison and analysis, the feeling is like a product evaluation not a technical paper. I would like to suggest that publication would be more appropriate in another journal.

Reviewer 2 Report

The authors have presented sound results on the topic. Here are my few concerns:

1. The introduction needs to be comprehensive, so do add details from the previous findings.

2. Discussion on the availability of full ground reaction force is required.

3. Estimation of the correlation between ground contact timing and running economy for the participants needs to be discussed 

4.  Few sentences appear to be complex, they need to be simplified into shorter ones 

Reviewer 3 Report

Thank you for the opportunity to review this interesting manuscript investigating the validity of a three-dimensional motion capture system and the Garmin Running Dynamics Pod for determining ground contact times while running in place by comparing them obtained with a bilateral force plate.

The article is well-written and clearly presents the obtained results. Please find my few comments below.

1.       In the introduction, a more comprehensive discussion of why measuring the running-in-place GCT and eventually how running-in-place characteristics were recorded in other investigations could be helpful to focus better on the importance of the presented investigation (now only one reference is provided).

2.       Data analysis: the authors used a ground reaction threshold of 20 N and applied the algorithm developed by Handsaker et al. (2016). However, Handsaker and colleagues used a threshold of 5 N that can lead to the recognition of a slightly anticipated Touchdown and a slightly delayed Toe-off compared with 20 N (therefore leading to a longer GCT). I was wondering if this difference in the methodological approach can partially explain the difference between the events measured with the platforms and MoCap.

3.       A note related to the fact that the in-built algorithm of the Garmin Running Dynamics Pod is developed to recognize the GCT during regular running could be added to the discussion.